A laboratory experiment on using different financial-incentivization schemes in software-engineering experimentation

http://orcid.org/0000-0002-9856-1548 Bershadskyy Dmitri 1 dmitri.bershadskyy@ovgu.de
Krüger Jacob 2 j.kruger@tue.nl
Calıklı Gül 3
http://orcid.org/0000-0001-9645-2586 Otto Siegmar 4
Zabel Sarah 4
Greif Jannik 1
Heyer Robert 5 6
1 Otto-von-Guericke Universität Magdeburg , Magdeburg , Germany
2 Eindhoven University of Technology , Eindhoven , Netherlands
3 University of Glasgow , Glasgow , United Kingdom
4 Universität Hohenheim , Stuttgart , Germany
5 Leibniz Institute for Analytical Sciences , Dortmund , Germany
6 Universität Bielefeld , Bielefeld , Germany
Wagner Stefan
Electronic publication date: 2025 Mar 12
Publication date: 2025
Volume: 11
Electronic Location ID: e2650
Received 2024 Oct 4; Accepted 2024 Dec 19
Copyright: © 2025 Bershadskyy et al.
Copyright year: 2025
Copyright holder: Bershadskyy et al.
License: This is an open access article distributed under the terms of the Creative Commons Attribution License, which permits unrestricted use, distribution, reproduction and adaptation in any medium and for any purpose provided that it is properly attributed. For attribution, the original author(s), title, publication source (PeerJ Computer Science) and either DOI or URL of the article must be cited.
License URL: https://creativecommons.org/licenses/by/4.0/

Keywords: Software engineering, Experimentation, Financial incentives

Funding: Otto-von-Guericke University Magdeburg, German Magdeburg University supported the APC The research reported in this article has been supported by the Innovation Fund of the Otto-von-Guericke University Magdeburg, Germany. The Open Access Publication Fund of Magdeburg University supported the APC. The funders had no role in study design, data collection and analysis, decision to publish, or preparation of the manuscript.

==============================
In software-engineering research, many empirical studies are conducted with open-source or industry developers. However, in contrast to other research communities like economics or psychology, only few experiments use financial incentives (i.e., paying money) as a strategy to motivate participants’ behavior and reward their performance. The most recent version of the SIGSOFT Empirical Standards mentions payouts only for increasing participation in surveys, but not for mimicking real-world motivations and behavior in experiments. Within this article, we report a controlled experiment in which we tackled this gap by studying how different financial incentivization schemes impact developers. For this purpose, we first conducted a survey on financial incentives used in the real-world, based on which we designed three incentivization schemes: (1) a performance-dependent scheme that employees prefer, (2) a scheme that is performance-independent, and (3) a scheme that mimics open-source development. Then, using a between-subject experimental design, we explored how these three schemes impact participants’ performance. Our findings indicate that the different schemes can impact participants’ performance in software-engineering experiments. Our results are not statistically significant, possibly due to small sample sizes and the consequent lack of statistical power, but with some notable trends that may inspire future hypothesis generation. Our contributions help understand the impact of financial incentives on participants in experiments as well as real-world scenarios, guiding researchers in designing experiments and organizations in compensating developers.

Motivation

Experimentation in software engineering rarely involves financial incentives to compensate and motivate participants. However, in most real-world situations it arguably matters whether software developers are compensated, for instance, in the form of wages or bug-bounties (Krüger, Nielebock & Heumüller, 2020; Krishnamurthy & Tripathi, 2006) of open-source communities. Particularly experimental economists use financial incentives during experiments for two reasons (Weimann & Brosig-Koch, 2019). First, financial incentives improve the validity of the experiment by mimicking real-world incentivisation schemes to motivate participants’ realistic behavior and performance. To this end, in addition to show-up or participation fees, the actual performance of participants during the experiment is rewarded by defining a payoff function that maps the participants’ performance during the experiment to financial rewards or penalties. Second, they allow to study different incentives with respect to their impact on participants’ performance. It is likely that using financial incentives in empirical software engineering can help improve the validity by mimicking and staying true to the real world, too.

Interestingly, there are no guidelines or recommendations on using financial incentives in software-engineering experimentation. For instance, the current SIGSOFT Empirical Standards (https://github.com/acmsigsoft/EmpiricalStandards) (Ralph, 2021), as of January 22, 2024 (commit 9374ea5), mention incentives solely in the context of longitudinal studies and rewarding participation in surveys to increase participation. Also, to the best of our knowledge and based on a literature review, financial incentives that reward participants’ performance during an experiment are not used systematically in empirical software engineering. Although some studies broadly incentivize performance (e.g., Sayagh et al. (2020) or Shargabi et al. (2020)), these do not aim to improve the validity of the experiment, only participation. Furthermore, we know from experimental economics (Charness & Kuhn, 2011; Carpenter & Huet-Vaughn, 2019) that finding a realistic (and thus externally valid) way to reward performance is challenging and no simple one-fits-all solution exists. For instance, the performance of open-source developers depends less on financial rewards than those of industrial developers (Baddoo, Hall & Jagielska, 2006; Ye & Kishida, 2003; Huang, Ford & Zimmermann, 2021; Beecham et al., 2008).

As a step towards understanding and systematizing the potential of using financial incentives in software engineering experimentation, we have conducted a two-part study comprising a survey and a controlled experiment in the context of bug detection through code reviews (Krüger et al., 2022). First, we used a survey with practitioners to elicit real-world incentivisation schemes on bug finding. In the survey, we distinguished between the schemes most participants prefer and those actually employed. Building on the results, we defined one payoff function for our experiment. Please note that we originally planned to have two functions from the survey, one for the most applied (MA) and one for the most preferred (MP) incentives (Krüger et al., 2022). However, the survey responses for the MA incentives were identical to no performance-based incentives, which we added as a control treatment anyway. To extend our experiment, we added two more payoff functions: one that is performance-independent and one that resembles the motives of open-source developers. We derived the latter function using the induced-value method established in experimental economics (Smith, 1976; Weimann & Brosig-Koch, 2019), which induces a controlled willingness of participants to achieve a desired goal (i.e., identify a bug) or obtain a certain good during an experiment by mimicking its monetary value (e.g., a reward). Second, we employed our actual between-subject experiment to explore to what extent each of the three payoff functions impacts the participants’ behavior. Overall, we primarily contribute to improving researchers’ understanding of whether and how financial incentives can help software engineering experimentation. However, our experiment can also help reveal whether different incentivisation schemes could improve practitioners’ motivation. Our survey and experimental design artifacts are available for peer-reviewing.

In total, we contribute the following in this article: 1. We find indications that different forms of financial incentives impact participants’ performance in software-engineering experiments. Due to the small sample sizes, our results are not statistically significant, but we still observe clear tendencies.

2. We discuss what our findings imply for using financial incentives in other software-engineering experiments, and for designing respective payoff functions.

3. We share our artifacts, including the design and results of our survey as well as experiment in anonymous form within a persistent open-access repository (https://osf.io/mcxed/).

Our findings can help researchers improve the validity of their software-engineering experiments by employing financial incentives, while also shedding light into how these can impact motivation in practice.

Related work

Experiments in software engineering are comparable to “real-effort experiments” in experimental economics, which involve participants who solve certain tasks to increase their payoffs. Consequently, we built on experiences from the field of experimental economics, which involves a large amount of literature on how and when to use financial incentives in real-effort experiments (van Dijk, Sonnemans & van Winden, 2001; Greiner, Ockenfels & Werner, 2011; Gill & Prowse, 2012; Erkal, Gangadharan & Koh, 2018). For instance, some findings indicate gender differences regarding the impact of incentivization schemes, which we have to consider during our experiment. In detail, research has shown that men choose more competitive schemes (e.g., tournaments, performance payments). Similarly, participants with higher social preferences select such competitive schemes more rarely (Niederle & Vesterlund, 2007; Dohmen & Falk, 2011). We considered such factors when analyzing the results of our experiment (e.g., comparing gender differences if the number of participants allows).

Unfortunately, there is much less research on incentivization schemes in software-engineering experimentation. Mason & Watts (2009) have analyzed the impact of financial incentives on crowd performance during software projects using online experiments. The results are similar to those in experimental economics, but the authors also acknowledge that they did not design the incentives to mimic the real world or to improve the participants’ motivation. Other studies have been concerned with the impact of payments on employees’ motivation (Sharp et al., 2009; Thatcher, Liu & Stepina, 2002), job satisfaction (Klenke & Kievit, 1992; Storey et al., 2021), or job change (Burn et al., 1994; Hasan et al., 2021; Graziotin & Fagerholm, 2019). For instance, Baddoo, Hall & Jagielska (2006) conducted a case study and found that developers perceived wages and benefits as an important motivator, but they did not connect payments to objective performance metrics. None of the studies we are aware of decomposed payments or wages into specific components (e.g., performance-dependent vs. performance-independent). So, the effectiveness of different payoff schemes on developers’ performance remains unclear.

Software-engineering researchers have investigated the motivations of open-source developers to a much greater extent (Gerosa et al., 2021; Hertel, Niedner & Herrmann, 2003; Hars & Ou, 2002; Ye & Kishida, 2003; Huang, Ford & Zimmermann, 2021). From the economics perspective, open-source systems represent a public good (Bitzer, Schrettl & Schröder, 2007; Lerner & Tirole, 2003): they are available to everyone and their consumption do not yield disadvantages to anyone else. A typical problem of public goods is that individual and group incentives collide, which usually leads to an insufficient provision of the good. While typical explanations for open-source development focus on high intrinsic motivation to contribute or learn, this is not always the case. For instance, Roberts, Hann & Slaughter (2006) show that financial incentives can actually improve open-source developers’ motivation (in terms of contributions). Still, financial incentives are at least not always the predominant motivators for software developers (Beecham et al., 2008; Sharp et al., 2009). As a consequence, we used the concept of open-source software as a social good (Huang, Ford & Zimmermann, 2021) as an extreme example (i.e., the developers help solve a social problem, but do not receive a payment) for designing one payoff function in our experiment.

Study protocol

As explained previously, our study involved two data-collection processes, a survey and a laboratory experiment. In Table 1, we provide an overview of our intended study goals based on the Peer Community In Registered Reports (PCI RR) (https://rr.peercommunityin.org/) study design template, which we explain in more detail in this section. Our study design was based on guidelines for using financial incentives in software-engineering experimentation (Krüger et al., 2024) and has received approval from the local Ethics Review Board of the Department for Mathematics and Computer Science at Eindhoven University of Technology, The Netherlands, on October 24, 2022 (reference ERB2022MCS21).

Table 1 PCI RR study design template for our initial study design. In the column deviations, we explain whether and why we deviated from this design (all changes were approved by the recommender).

Question	Hypothesis	Sampling plan	Analysis plan	Sensitivity rationale	Interpretation	Disproved theory	Deviations	Observed outcome	
Which payoff functions are applied/preferred in SE practice? (survey)	N/A	At least 30 participants (personal contacts and social media).	We analyzed the absolute frequency of the combinations of payment components. We computed the mean values of the weights for the MA and MP combinations.	N/A	If MAIT and MPIT were identical, we would have reduce the number of treatments from four to three.	N/A	We conducted an additional iteration of the (translated) survey with eight participants from a German company to achieve our anticipated sample size.	The most commonly applied payments are fixed. The most commonly preferred one is a combination of fixed payment and company-performance-dependent bonus.	
How do different payoff functions impact the performance of participants in SE experiments? (experiment)	H1: Participants without performance-based incentivization (NPIT) have on average a worse performance than those with performance-based incentivization (e.g., OSIT, MAIT, MPIT).	We aimed to recruit at least 80 (20 per treatment) computer-science students of the Otto-von-Guericke University Magdeburg. Furthermore, we conducted an a posteriori power analysis to reason on the power of our tests.	If their assumptions were fulfilled, we used parametric tests to compare between the treatments. Otherwise, we employed non-parametric tests. For H1, we used pairwise comparisons of the performance-independent treatment to the other treatments: NPIT vs. MPIT

NPIT vs. MAIT

NPIT vs. OSIT

	Due to our experimental design, we faced the issue of multiple hypotheses testing. We addressed this issue by applying the Holm-Bonferroni correction.	We find support for H1, if our participants’ performance in NPIT is significantly lower than in any other of our experimental treatments at p<0.05—after correcting with the Holm-Bonferroni method: (NPIT< MPIT) OR (NPIT< MAIT) OR (NPIT< OSIT). Confirming H1 means that the performance is better in the specific treatment with performance-based incentives compared to NPIT. This implies that if performance plays a role in a software-engineering experiment, performance-based incentivization should be considered.	There is no theory focusing on the role of incentives in software engineering. Incentivization in software-engineering experiments is scarcely applied. Our results can improve experimental designs in software engineering by guiding researchers when and how to use incentives in their experiments.	While we anticipated the possibility that MAIT and MPIT would be identical and should be merged, this did not happen. However, we found that MAIT and NPIT were essentially identical, which is why we merged these two. The changes were made prior to the commencement of the experiment and were approved by PCI RR on 06 Dec 2022.	The results of the pre-registered tests were non-significant. Yet, they indicate notable differences that guided our exploratory analysis.	
	H2: The experimental performance of participants under performance-based incentivization (e.g., OSIT, MAIT, MPIT) differs between treatments.		For H2, we used pairwise comparisons of the performance-dependent treatments: MPIT vs. MAIT

MAIT vs. OSIT

OSIT vs. MPIT

		We find support for H2, if our participants’ performance between the treatments differs and the respective tests are significant with p<0.05—after correcting with the Holm-Bonferroni method: (MPIT<> MAIT) OR (MAIT<> OSIT) OR (OSIT<> MPIT). Confirming H2 means that the practitioners’ performance differs depending on the type of incentivization. If we cannot confirm H2, we do not find evidence for OSIT, MAIT, and MPIT to induce different performances.				
In total, we would compute up to six pairwise tests to compare the (at most) four treatments with one another and corrected for multiple hypotheses testing (Holm-Bonferroni method). We also conducted regression analyses using the treatments as categorical variables (NPIT as base) and age, gender, experience, as well as arousal as exogenous variables	
Note:

NPIT, No Performance Incentives Treatment; OSIT, Open-Source Incentives Treatment; MAIT, Most-Applied Incentives Treatment; MPIT, Most-Preferred Incentives Treatment.

Survey design

Goal

With our survey, we aimed to explore i) which payment components (e.g., wages only, bug bounties) are most applied (MA) in practice and ii) which payment components are most preferred (MP) by practitioners. We display an overview of these payment components with concrete examples in Table 2. Our intention was to understand what is actually employed compared to what would be preferred as a payment schema to guide the design of our experiment.

Table 2 List of components of payment we asked about in our survey to design payoff functions for the experiment.

Note that the term check refers to participants selecting or deselecting a line of code during our experiment (i.e., marking them as buggy or correct as can be seen in Fig. 1).

Payment component	Example	Variable	
Not performance-based	
Hourly wage	Payment for hours spent on code review	wage	
Payment per task	Fixed payment for conducting a code review	paymentfix	
Others	Specified by participants		
Performance-based	
Reward for completing review	Bonus for finding all bugs	rewardcomplete	
Reward for quality	Bonus for correctly found bug (e.g., bug bounty)	rewardquality	
Reward for time	Bonus for performing reviews fast	rewardtime	
Reward for organization’s performance	Bonus provided based on the organization’s profits	rewardshare	
Penalty for low quality	Penalty for mistakes within a certain period (e.g., missed bugs)	penaltyquality	
Penalty for checks	Penalty for marking lines of code in the experiment	penaltycheck	
Penalty for required overtime	Penalty for not completing within working hours	penaltytime	
Others	Specified by participants		

Figure 1 Screenshot of the code example as we showed it to the participants. The checkboxes in front of each line allowed the participants to check buggy lines of code.

Note that we did not show the comments indicating the implemented bugs (i.e., in lines 16, 21, and 38). The blue boxes (not displayed to participants) indicate the Areas of Interest (AOIs) that we used for the eye-tracking analysis.

Structure

To achieve our goal, we created an online questionnaire with the following structure (cf. Table 3). At first, we welcomed our participants, informing them about the survey’s topic, duration, and their right to withdraw from our experiment at any point in time without any disadvantages. Furthermore, we asked for written consent to collect, process, and publish the data in anonymized form. To allow for questions, we provided the contact data of one author on the first page. Then, we asked about each participant’s background to collect control variables, for instance, regarding their demographics, role in their organization, the domain they work in, and experience with code reviews. These background questions allow us to monitor whether we have acquired a broad sample of responses from different organizations, and thus on varying practices. Our goal was to mitigate any bias caused by external variables, such as the organizations’ culture. Also, we discarded the answers of one participant who had no experience with code reviews. Based on the participants’ roles, the online survey showed the questions on the payment structures in an adaptive manner. We designed these questions as well as their answering options based on established guidelines and our experiences with empirical studies in software engineering (Siegmund et al., 2014; Nielebock et al., 2019; Krüger et al., 2019).

Table 3 List of variables we checked in our survey.

Variable	Description	Operationalization	
Control variables	
Demographics	Age, gender, living country, highest level of education	Nominal (single-choice list)	
Role	Participant’s role in their organization	Nominal (single-choice list)	
Experience	Years of experience in software development and code reviewing	6-level Likert scale (<1 – >15)	
Frequency	Current involvement in software development and code reviewing	5-level Likert scale (none at all – daily)	
Domain	Domain of the participant’s organization	nominal (single-choice list)	
Size of organization	Number of employees	5-level Likert scale (<21 – >200)	
Size of team	Number of members in participant’s team (if applicable)	6-level Likert scale (1 – >50)	
Development process	Whether agile or traditional development processes are employed	° agile ° non-agile	
Target variables	
MA/MP incentives	List of payment components that can be selected (cf. Table 2)	Nominal (checklist)	
MA/MP percentage	Percentage to weigh the payment components chosen before	Continuous (0–100%)	
Working hours per week	Weekly working hours according to the participant’s contract	Continuous	
Unpaid overtime	Potential unpaid overtime of employees in proportion to working hours per week	Ratio	
Note:

MA, most applied; MP, most preferred.

To explore the payment components (target variables), we displayed the ones we summarize in Table 2. We used a checklist in which a participant could select all components that are applied in their organization. Each selected component had a field in which the participant could enter a percentage to indicate to what extent that component impacted their payment (e.g., 80% wage and 20% bug bounty). Then, we presented the same checklist and fields again. This time, the participant should define which subset of the components they would prefer to contribute with what share to the payment. While we presented this second list as is to any management role (e.g., project manager, CEO), we asked software engineers (e.g., developer, tester) to decide upon those components from the perspective of being the team or organization lead. To prevent sequence effects, we randomized the order in which the two treatment questions occurred (applied and preferred). Finally, we asked each participant to indicate how many hours per week they worked unpaid overtime—which represents a type of performance penalty for our payoff functions—and allowed them to enter any additional comments on the survey.

Sampling participants

We invited personal contacts and collaborators from different organizations, involving software developers, project managers, and company managers. Note that we excluded self-employed or freelancer developers who typically ask for a fixed payment for a specific task or project. In addition, we distributed a second version (to distinguish both populations) of our survey through our social media networks. In consultation with the PCI Recommender (December 6, 2022), we surveyed an additional sample of eight employees from a company to obtain a larger sample size. For this additional sample, we translated the questionnaire into German. We tested whether there are differences between the samples regarding our variables of interest. If the MA and MP incentives were identical in all samples, we would have collapsed the data. Otherwise, we would have built on the sample of our personal contacts only. This allowed us to have a higher level of control over the participants’ software-engineering background, and their experience with code reviews.

Our goal was to acquire at least 30 responses to obtain a reasonable understanding of applied and preferred payments. Since we did not evaluate the survey data using inferential statistics, we based our sample-size planning on the limited access to a small, specialized number of potential participants. Note that we did not pay incentives for participating in the survey. We expected that the survey would take 10 min at most, and did verify the required time and understandability of the survey through test runs with three PhD students from our work groups.

Analysis plan

To specify the payoff functions for our experiment, we considered the absolute frequency of combinations of different payment components. Precisely, to identify the MA and MP combinations, we chose the respective combination that was selected by the largest number of respondents (i.e., modal value). For these two combinations, we computeed the mean values for their weights. We performed a graphical-outlier analysis using boxplots (Tukey, 1977), excluding participants with extreme values (i.e., three inter quartile ranges above the third quartile or below the first quartile). As an example, assume that most of our participants would state to prefer the combination of fixed wages (with a weight of 75% on average) and bug bounties (25% on average). Then, we would define a cost function as 0.75⋅paymentfix+0.25⋅(bugscorrect⋅rewardquality).

Threats to validity

Our survey relied mostly on our personal contacts, which may have biased its outcomes. We mitigated this threat, since we have a broad set of collaborators in different countries and organizations. Moreover, defining the “ideal” payoff function for practitioners may pressure the participants, is hard to define (e.g., considering different countries, organizational cultures, open-source communities, or expectations), and challenging to measure (e.g., what is preferred or efficient). However, this is due to the nature of our experiment and the property we study: financial incentives. Consequently, these threats remain and we discuss their potential impact, which can only be mitigated with an appropriately large sample population.

Laboratory experiment

Goal

After eliciting which payoff functions are used and preferred in practice, we conducted our actual experiment to measure the impact of different payoff functions in software-engineering experiments. We focused on code reviews and bug identification in this experiment, since these are typical tasks in software engineering that also involve different types of incentives. So, we aimed to support software-engineering researchers by identifying which payoff functions can be used to improve the validity of experiments.

Treatments

As motivated, we aimed to compare four treatments to reflect different payoff functions that stemmed from our survey and established research. While we were able to define the payoff functions for the “No Performance Incentives Treatment” (NPIT) and “Open-Source Incentives Treatment” (OSIT) in advance, we needed data from our survey to proceed with the “MP Incentives Treatment” (MPIT) and “MA Incentives Treatment” (MAIT). However, we did a priori describe the method we would use to derive the payoff functions for those treatments. Note that some treatments could yield the same payoff function (i.e., NPIT, MAIT, and MPIT). It is unlikely that all three payoff functions would be identical, but we merged those that were (i.e., NPIT and MAIT) and reduced the number of treatments accordingly (see Table 2 for the variable names):No Performance Incentives Treatment (NPIT): For NPIT, we provided a fixed payment (i.e., 10 €) that was payed out at the end of an experimental session. So, this treatment mimics a participation fee for experiments or fixed wages for the real world. Consequently, the payoff is independent of a participant’s actual performance. Overall, the payoff function (PF) for this treatment is:

PFNPIT=paymentfix.

Open-Source Incentives Treatment (OSIT): Again, this treatment does not depend on our survey results, but builds on findings from the software-engineering literature on the motivation of open-source developers (Gerosa et al., 2021; Hertel, Niedner & Herrmann, 2003; Hars & Ou, 2002; Ye & Kishida, 2003; Huang, Ford & Zimmermann, 2021). We remark that we focused particularly on those developers that do not receive payments (e.g., as wages or bug bounties), but work for free. In a simplified, economics perspective, such developers still act within a conceptual cost-benefit framework (i.e., they perceive to obtain a benefit from working on the software). We built on the induced-value method (Weimann & Brosig-Koch, 2019) from experimental economics to mimic this cost-benefit framework with financial incentives to derive the OSIT treatment. Besides a participation fee, we involved a performance-based reward for correctly identifying all bugs to resemble goal-oriented incentives (e.g., personal fulfillment of achieving a goal or supporting open-source projects). Furthermore, we considered the opportunity costs of working on open-source software (i.e., less time for other projects and additional effort for performing a number of checks). Overall, the payoff function (PF) for this treatment is:

PFOSIT=paymentfix+rewardcomplete−time⋅penaltytime−checks⋅penaltychecks.

MA Incentives Treatment (MAIT): Using our survey results, we could identify a payoff function that represents what is mostly applied in practice. We would then derive a payoff function as explained in “Survey Design”. However, we found that the survey results led to the same function as for NPIT, which is why we did not use a distinct MAIT in our actual experiment.

MP Incentives Treatment (MPIT): We used the same method we used for MAIT to define a payoff function for MPIT. In this case, the developers preferred a fixed payment with an additional quality reward that is based on their organization’s performance:

PFMPIT=paymentfix+rewardquality⋅rewardshare.

Note that these payoff functions cannot be perfect, but they are mimicking real-world scenarios, and thus are feasible to achieve our goals.

We used the same code-review example for all treatments to keep the complexity of the problem constant. For all treatments, we calibrated the payoff function so that the expected payoff for each participant in and between treatments was approximately the same (i.e., around 10 €). Implementing similar expected payoffs avoids unfairness between treatments, and ensures that performance differences are caused by different incentive schemes and not the total size of the payoff. After the treatment, we gathered demographic data from the participants (e.g., age, gender) and asked for any concerns or feedback. We estimated that each session of the experiment would take 45 min.

Code example

We selected and adapted three different Java code examples (i.e., limited calculator, sorting and searching, a Stack), the first from the learning platform LeetCode (https://leetcode.com) and the other two from the “The Algorithms” GitHub repository (https://github.com/TheAlgorithms/Java). To create buggy examples, we injected three bugs into each code example by using mutation operators (Jia & Harman, 2011). Note that we partly reworked the examples to make them more suitable for our experiment (e.g., combining searching and sorting), added comments at the top of each example explaining its general purpose, and kept other comments (potentially adapted) as well as identifier names to improve the realism. We aimed to limit the time of the experiment to avoid fatigue and actually allow for a laboratory setting, and thus decided to use only one example. To select the most suitable subject system for our experiment, we performed a pilot study in which we measured the time and performance of the participants. In detail, we asked one M.Sc. student from the University of Glasgow who has worked as a software practitioner in industry and four PhD students from the University of Zurich to perform the code reviews on the buggy examples. Overall, each example was reviewed by three of these participants. Our results indicated that the sorting and searching example would be most feasible (i.e., ≈12 min., 4/9 bugs correctly identified, Five false positives), considering that the task should neither be too easy nor to hard, the background of the pilot’s participants and the potential participants for our experiment, as well as the examples’ quality. The other two examples seemed too large or complicated (i.e., ≈14 min., 2/9 bugs; four false positives; ≈8 min., 5/9 bugs, eight false positives), which is why we decided to use the sorting and searching example (available in our artifacts) (https://osf.io/mcxed/). We remark that none of the participants from this pilot study was involved in our actual experiment. In Fig. 1, we display a screenshot of the sorting and searching code example we showed to the participants in the lab.

Sampling participants

We aimed to recruit a minimum of 80 participants (20 per treatment) by inviting students and faculty members of the Faculty for Computer Science of the Otto-von-Guericke University Magdeburg, Germany. In 2019, 1,676 Bachelor and Master students as well as roughly 200 PhD students had been enrolled at the faculty, and 193 (former) members of the faculty were listed in the participant pool of the MaXLab (https://maxlab.ovgu.de/en/) at which we conducted the laboratory experiment. We focused on recruiting participants who passed the faculty courses on Java and algorithms (first two semester) or equivalent courses to ensure that our participants had the fundamental knowledge required for understanding our sorting and searching example. If possible (e.g., considering finances, response rate), we planned to invite further participants (potentially from industry and other faculties) to strengthen the validity of our results. Yet, it was not realistic to have more than 35 participants per treatment, due to the number of possible participants with the required background on software engineering. Applying the Holm-Bonferroni correction for multiple hypothesis testing, we calculated the power analysis for the strictest corrected α of 0.0083 ( 0.05/6) in the range between 20 and 35 participants per treatment. A Wilcoxon-Mann-Whitney test for independent samples with 20/35 participants per group (N = 40/70) would be sensitive to effects of d=1.33/1.08 with 90% power ( α=0.0083). This means that our experiment would not be feasible to reliably detect effects smaller than Cohen’s d=1.33/1.08 within the range of realistic sample sizes. In Fig. 2, we illustrate this relation between effect and sample size. Overall, it was unlikely that we would identify statistically significant differences. Note that we focused on the Otto-von-Guericke University, since the MaXLab is located there. Regarding the Covid pandemic, it was possible to conduct sessions only with reduced numbers of participants (i.e., 10 instead of 20). We were not aware of any theory or previous experiments on the effect of financial incentives on developers’ performance during code reviews or other software-engineering activities. As a consequence, we could not confidentially define what the smallest effect size of interest would be.

Figure 2 Relation between sample size and Cohen’s d for comparing two groups via the Wilcoxon- Mann-Whitney test, assuming a normal distribution with α=0.0083 and statistical power of 0.9.

Hypotheses

Reflecting on findings in software engineering as well as other domains, we defined two hypotheses (H) we wanted to study in our experiment:

H1 Participants without performance-based incentivization (NPIT) have on average a worse performance (lower value in the F1-score, explained shortly) than those with performance-based incentivization (e.g., OSIT, MAIT, MPIT).

H2 The experimental performance of participants under performance-based incentivization (e.g., OSIT, MAIT, MPIT) differs between treatments.

Besides analyzing these hypotheses, we also compared the behavior (e.g., risk taking) and performance between all groups to understand which incentives have what impact. Moreover, we used eye trackers to explore fixation counts, fixation lengths, and return fixations. This allowed us to obtain a deeper understanding of the search and evaluation processes during code reviews. Also, it enabled us to investigate potential differences in eye movements depending on the incentivization. More precisely, we intended to follow similar studies from software engineering (Abid et al., 2019) to explore how our participants read the source code, for instance, did they focus on the actually buggy code, what lines were they reading more often, or which code elements did they focus on to explore bugs? We report all findings from the eye-tracking data as exploratory outcomes. The eye-tracking data is preprocessed by the firmware of Tobii (Version 1.181.37603) using the Tobii I-VT (fixation) filter.

Metrics

The performance of our participants was primarily depending on their correctness in identifying bugs during the code review. Since this can be expressed as confusion matrices, we decided to implement the F1-score (definedas2TP2TP+FP+FN) as the only outcome measure to evaluate our hypotheses. For our experiment, true positives (TP) refer to the correctly identified bugs, false positives (FP) refer to the locations marked as buggy that are actually correct, and false negatives (FN) refer to the undetected bugs. Note that our participants were not ware of this metric (they only knew about the payoff function) to avoid biases, and any decision based on the payoff function are reflected by the F1-score (e.g., taking more risks due to missing penalties under NPIT). So, this metric allowed us to compare the performances of our participants between treatments considering that they motivate different behaviors, which allowed us to test our hypotheses. In summary, our dependent variable was the F1-score, our independent variable was the payoff function, and we collected additional data via a post experimental survey (e.g., experience, gender, age, stress) as well as eye-tracking data for exploratory analyses.

Experimental design

Fundamentally, we planned to employ a 4 × 1 design. However, since we merged the treatments NPIT and MAIT after our survey, we ended up with a 3 × 1 design). For each treatment, we only changed the payoff function. We allocated participants to their treatment at random, without anyone repeating the experiment in another treatment. On-site, we could execute the experiment at the experimental laboratory MaXLab of the Otto-von-Guericke University using a standardized experimental environment. We employed a between-subject design measuring the participants’ performance and measured the eye movement of four participants (restricted by number of devices) in each session using eye trackers (60 Hz Tobii Pro Nano H). Note that we could identify any impact wearing eye-trackers may have had on our participants during our analysis. However, it is not likely that they had an impact, because this type of eye trackers is mounted to the screen and barely noticeable, not a helmet the participants have to wear. The procedure for each session was as follows:Welcome and experimental instructions: After the participants of a session entered the laboratory, they were randomly allocated to working stations with the experimental environment installed. Moreover, four of them were randomly selected for using eye trackers. To this end, we already stated in the invitation that eye tracking would be involved in the experiment. If a participant nonetheless disagreed to participate using eye trackers, we excluded them from the experiment to avoid selection bias. Once all participants were at their places, the experimenter began the experiment. The participants received general information about the experiment (e.g., welcoming text), information about the task at hand (code review), an explanation on how to enter data (e.g., check box), and the definition of their payoff function for the experiment (with some examples).

Review task: All participants received the code example with the task to identify any bugs within it. Note that the participants were not aware of the precise number of bugs in the code. Instead, a message explained that the code does not behave as expected when it is executed. At the end of the task, we could have incorporated unpaid overtime as a payment component by asking participants to stay for five more minutes to work on the task.

Post experimental questionnaire: After the experiment, we asked our participants a number of demographic questions (i.e., gender, age, study program, study term, programming experience). We further applied the distress subscale of the Short Stress State Questionnaire (Helton, 2004) to measure arousal and stress of the participants. Eliciting such data on demographics and arousal enabled us to identify potential confounding parameters.

Payoff procedure: After we collected all the data, we provided information about their performance and payoff to the participants by displaying them on their screen. We payed out these earnings privately in a separate room in cash immediately afterwards.

Analysis plan

To analyze our data, we employed the following steps:Data cleaning: The experimental environment stored raw data in CSV files. We did not plan to remove any outliers or data unless we would identify a specific reason for which we would believe the data could be invalid, which involved primarily two cases. First, it may have happened that the eye-movement recordings of a participant have a low quality (i.e.,<70% gaze sample). Gaze sample is defined as the percentage of the time that the eyes are correctly detected. Since we used eye tracking only for exploratory analyses, we would not have replaced participants just because the calibration was not good enough. Moreover, the participants were not aware of the quality and could simply continue with the actual experiment. However, we excluded their eye-tracking data from our exploratory analysis. Second, we would have excluded participants if they violated the terms of conduct of the laboratory. While this case was unlikely, we would have tried to replace these participants to achieve the desired sample sizes before data cleaning. Fortunately, neither of such cases occured.

Descriptive statistics: We used descriptive statistics for the demographic, dependent, and independent variables for each treatment by, reporting means and standard deviations of the respective variables.

Observational descriptions: Since sole statistical testing is often subject to misinterpretation and not recommended (Wasserstein & Lazar, 2016; Wasserstein, Schirm & Lazar, 2019; Amrhein, Greenland & McShane, 2019), we focused on describing our observations. For this purpose, we started with reporting the results we obtained, plotting suitable visualizations, and identifying patterns within these. The statistical tests helped us to improve our confidence in these observations.

Inferential statistics: For our analysis, we focused on performance (i.e., F1-score). We first checked whether the assumptions required for parametric tests (e.g., normality) are fulfilled, and if not proceeded with non-parametric tests (i.e., Wilcoxon-Mann-Whitney test). Since we were interested in all possible differences between the three treatments, we had to conduct all pairwise treatment tests. For the significance analyses, we applied a significance level of p<0.05 and corrected for multiple hypotheses testing using the Holm-Bonferroni method. Although the share of participants who used eye trackers was constant among all treatments, and thus should not affect treatment effects, we further checked whether the presence of eye trackers affected performance. To increase the statistical robustness, we also conducted a regression analysis using the treatments as categorical variables and NPIT as base. As exogenous variables, we included: age, gender, experience, and arousal of the participants. In contrast to the preregistered tests, we discuss these results as exploratory outcomes.

Based on these steps, we obtained a detailed understanding of how different incetivization schemes impact the performance of software developers during code review.

Results

In this section, we first report the results of our survey that we used to motivate the incentive structures in our experiment, and then the results from the experiment itself.

Survey

In line with our Stage 1 registered report (Krüger et al., 2022), we obtained a total of 39 responses to our survey. After excluding those respondents who did not provide responses for MAIT or MPIT, the final sample size was 30 respondents. Before we proceeded, we first checked whether the MAIT and MPIT were identical in all three sub-samples (personal contacts, social media, contacted company). We found that the components for MAIT were identical across all three samples. For MPIT, we identified a tie in the social media and the company samples between the combination “monthly fixed salary + company bonus” and “monthly fixed salary only.” Yet, in the personal contacts sample, the combination of fixed salary and company bonus was the sole first rank. Due to the small sample size, significance tests for differences in the samples are not meaningful. Therefore, we decided that it would be useful to pool all three sub-samples. We display the absolute frequencies of the payment components in the survey in Table 4. Based on the responses, we selected the two combinations (MAIT and MPIT) that were most frequently chosen by the participants. Note that, particularly with regard to the desired payment components, many different combinations were chosen from the components listed in the survey. We only took the most frequently selected combinations into account. Therefore, the following numbers differ from the absolute frequency of the selected components in Table 4.

Table 4 Comparison of the MA and MP payment components.

Payment components	MA	MP	
Hourly wage (payment for hours spent on a task)	24	16	
Payment per task (fixed payment for conducting a task, independent of the duration, e.g., freelancers)	2	0	
Bonus for completing a task (e.g., finding all bugs)	0	3	
Bonus for quality of own work (e.g., for each correctly identified bug)	0	12	
Bonus for performing tasks fast	0	9	
Bonus linked to company performance	12	16	
Malus for low quality (penalty for mistakes within a certain period, e.g., missed bugs)	0	0	
Malus for slow work (penalty for spending too much time on a task)	0	0	
Mean overtime (hours)	1.34	0.62	
Others (please indicate)	1	1	
Note:

The values represent absolute frequencies, except for “overtime,” which is measured in hours.

We derived the following from our survey results. Regarding the MA combination, 15 respondents indicated receiving only an hourly or monthly fixed wage. The second most frequently applied combination in our sample was a fixed wage plus a bonus for company performance (six). The remaining participants stated various other combinations, for instance, task-related payment (two) or a combination of fixed wage plus a bonus for their own performance. Based on this, the MAIT should also be a fixed payment, which means that the incentive scheme would be the same as in NPIT. Therefore, we decided to merge these two groups in our experiment. In contrast, the MP incentive components were a combination of a fixed wage and a company-performance-based bonus (seven). The second most preferred payment scheme was a fixed wage only (six), followed by different other combinations, such as a bonus for the quality of own work accompanied by a bonus for company performance (two). The most preferred combination (i.e., fixed wage plus company performance) was stated by seven respondents, with five of them also defining their preferred mix of shares of fixed wage and company bonuses. The mean value of this preferred share is 83% for fixed wage and 17% for company bonus. This means that the fixed wage should be the major component of the total wage. We used this information to calculate the payoff function for MAIT in our experiment.

To summarize, mostly fixed payments and bonuses are applied in practice. However, our participants would also like good performance to be represented in payoffs, for instance, regarding the company’s success or the quality of their own work.

Finally, we present the demographics of our survey respondents in Table 5. The mean age of the respondents was 37.20 years (standard deviation: 8.32 years) and three were female. Our respondents indicated that they worked for 38.64 h per week on average (standard deviation: 4.54 h), and the majority (17) was employed in larger companies with a minimum of 200 employees. Most of our respondents were programmers (12), worked in Germany (20), and used agile methods (25). The experience in programming among the respondents varied, ranging from less than a year to over 10 years, with the frequency of programming ranging from once a month to daily. Regarding the educational background, our respondents had a wide range of different degrees. There was one respondent who stated that they had no experience in code reviews. We did not include the answers of this respondent regarding MAIT and MPIT in our analysis (yet, its inclusion would not have changed the results).

Table 5 Overview of the 30 survey respondents’ demographics.

Variable	Value	Responses	
Company size (employees)	>200	17	
	100–200	10	
	20–50	2	
	1–20	1	
Role	Programmer/Developer	12	
	Project lead	4	
	Software architect	4	
	Manager	3	
	Researcher	2	
	Tester	2	
	Consultant	1	
	IT staff	1	
	Product owner	1	
Country	Germany	20	
	n/a	3	
	Turkey	3	
	Sweden	2	
	Switzerland	1	
	United Kingdom	1	
Project management process	Agile	25	
	Non-agile	4	
	n/a	1	
Programming experience (years)	<1	1	
	1–2	2	
	>2–5	4	
	>5–10	10	
	>10	9	
	n/a	4	
Frequency of programming	Not at all	2	
	About once a month	6	
	About once a week	4	
	About once a day or more often	15	
	n/a	3	
Education	College/2-year degree or equivalent	1	
	Bachelor in computer science	5	
	Bachelor in STEM	1	
	Master in computer science	9	
	Master in STEM	4	
	PhD or higher title in computer science	3	
	PhD or higher title in STEM	2	
	n/a	6	

Experiment

Preregistration analysis

Due to the results of our preregistered survey, we implemented only three treatments instead of the originally planned four, since MAIT and NPIT turned out to be the same in terms of the components involved. In line with the methods for incentivization from experimental economics by Smith (1976), we designed three payoff functions that fulfill the criteria of salience, monotonicity, and dominance. This means that all subjects knew a priori how their payoff depends on their behavior in the experiment (salience), the chosen incentive (i.e., money) is better whenever there is more of it (monotonicity), and the total size of the expected payoff is high enough to dominate other motives of behavior like boredom (dominance). Overall, we derived the following concrete values for our three payoff functions (see “Laboratory Experiment” for the respective variables).

For MPIT, we used the information from our survey that suggested an 83% to 17% proportion between fixed and team-dependent-bonus payment to be preferred by our respondents. As a team we considered groups of more than two participants in MPIT within an experimental session. All participants were saliently informed that their payoff will depend on the average performance of the other participants in their session (salience). We approximated this proportion between fixed and team-dependent-bonus by making the average number of bugs found in a team within a session contribute an additional 10% of the fixed payment. Concretely, with the fixed amount of 25.00 €, participants received an additional x⋅2.50€ whenever the team found x bugs on average. This means, that when participants within a team find on average two bugs out of three, we are very close to the preferred allocation of fixed and performance-dependent components.

For OSIT, we used the induced value method (Smith, 1976). Our main assumption for the payoff function was that for open-source developers, finishing their open-source project (or a task therein) is highly valuable. We implemented this assumption by offering a very high bonus if all bugs were found correctly (i.e., goal achieved). However, open-source developers’ motivation does not depend solely on task fulfillment, meaning that there should be a performance-independent component. Also, working on a project costs time that could be spent otherwise (e.g., on the job or other projects). We implemented these two assumptions through a fixed payment and by subtracting money per time unit spent in the experiment. The reduction per time unit should not be too high, as we were not aware of any prior literature indicating how to balance this component. Yet, it is necessary to approximate this continuous decision of open-source developers. Finally, we implemented a penalty for submitting marked lines of code for two reasons: First, this penalty mimics the real world where thinking that something is a bug that is not, costs time (e.g., looking for unnecessary solutions). Second, the penalty ensures that it is less attractive for participants to simply mark all lines of code, since doing so would mean they will find all bugs and get the bonus. Therefore, the size of this penalty has to be considered jointly with the size of the payoff for finding all bugs.

For NPIT, there was only a fixed amount of money for taking part in the experiment. Finally, these considerations raised the question of how high the payoffs had to be to be dominant, while the average expected payoff should be similar across all treatments (i.e., (30 €). We drew estimates on which and how many bugs would be found in what time from our pilot experiment (cf. “Laboratory Experiment”). In our case this led to the following payoff functions:

(1) PFNPIT=30€

(2) PFMPIT=25€+2.5€bug⋅averagenumberofbugsfoundinteam

(3) PFOSIT=20€+30€ifallbugsfound−min.spent⋅0.2€min.−checksdone⋅1€check.

In the following, we first present the descriptive statistics for our treatments (cf. Table 6). For our confirmatory analysis, we did not have to exclude any participants from our experiment. Following the preregistered analysis plan, we disclose that out of 31 participants with eye-tracking devices, we had to exclude seven for our exploratory analysis due to either insufficient gaze detection or insufficient calibration results. Since these participants’ remaining data was still valid, we removed only their data for the exploratory eye-tracking analysis. Unfortunately, we did not achieve our goal of 30 participants per treatment, but only 22 to 23. While this meant less statistical strength, we nonetheless obtained important insights into the participants’ behavior.

Table 6 Descriptive summary of the participants in each treatment.

	NPIT	OSIT	MPIT	
Average age	23.59	25.00	25.04	
Male/Female/Diverse	17/5/0	18/4/0	16/7/0	
Programming years	4.46	3.82	4.00	
Study duration	4.86	3.96	7.39	
Programming courses	4.41	3.32	3.91	
Programming experience	5.82	5.68	5.00	
Number of participants	22	22	23	
Among these with eye-tracking	10	9	12	

According to our registered report, we focused on the F1-score as the measure of participants’ performance. As our experimental data does not fulfill the assumptions for a parametric test (Shapiro-Wilk test, NPIT: p-value < 0.01, OSIT: p-value < 0.01, MPIT: p-value < 0.01), we proceeded with the Wilcoxon-Mann-Whitney test for our statistical tests. Adjusted p-values (padjusted) stem from the Holm-Bonferroni correction. To investigate H1 (cf. Table 1), we compared NPIT with OSIT and MPIT, respectively. Despite the notable differences in the F1-scores (0.26 vs. 0.16 and 0.15), our statistical tests indicate no significant result (NPIT-OSIT: p-value = 0.896, padjusted > 0.99), NPIT-MPIT: p-value = 0.923, padjusted > 0.99), which is in large part due to our hypothesis stating that participants would perform better when performance incentives are in place. Instead, we see indications for the opposite. This is a surprising result, and we will provide some insights on possible explanations in the exploratory analysis. With respect to the two performance-dependent treatments (MPIT, OSIT), we also see no significant differences with respect to the F1-score (p-value = 0.796, padjusted > 0.99).

As the last step of our preregistered analysis plan, we conducted a regression analysis. The results of the Tobit regression with limits at 0 and 1 (cf. Table 7) mostly confirm our previous findings (performance in NPIT is non-significantly better than in OSIT and MPIT). Yet, adding a parameter (completion Time) that we did not preregister in model (3) indicates the importance of the completion time on the F1-scores. The longer the participants stayed in the experiment, the higher was their F1-score. We will address the topic of completion time in more detail in the following exploratory analysis.

Table 7 Results of the Tobit regression analysis.

	Dependent variable:	
	F1	
	(1)	(2)	Exploratory (3)	
TreatmentOSIT	−0.171 (0.132)	−0.144 (0.138)	−0.054 (0.136)	
TreatmentMPIT	−0.134 (0.128)	−0.146 (0.137)	−0.208 (0.134)	
Age		−0.004 (0.014)	−0.010 (0.013)	
GenderWoman		0.176 (0.122)	0.175 (0.116)	
ProgrammingExperience		−0.003 (0.034)	−0.016 (0.033)	
Engagement		0.018 (0.043)	0.042 (0.042)	
Distress		−0.042 (0.048)	−0.060 (0.046)	
Worry		0.005 (0.042)	−0.005 (0.041)	
CompletionTime			0.016* (0.006)	
LogSigma	−0.927** (0.141)	−0.955** (0.141)	−1.012** (0.140)	
Constant	0.139 (0.094)	0.213 (0.417)	0.144 (0.399)	
Notes:

* p < 0.05.

** p < 0.01.

Exploratory analysis

As we had to decide on one specific variable to measure performance, we chose the F1-score—because it balances the different types of correct and wrong assessments. However, this decision is usually made with respect to the severity of different types of errors, for instance, a false negative and false positive need not be of equal importance for the company. Therefore, we now display the differences in treatments for all four categories: true positives (TP), true negatives (TN), false positives (FP), and false negatives (FN). As we can see in Fig. 3, this data indicates substantial differences between some of the metrics. For example, participants in OSIT had a low value of TP and a high value of FN ( x¯TP=0.59, x¯FN=2.41).

Figure 3 Boxplots for TP, TN, FP, and FN across our treatments.

Each box shows the 25% and 75% quantiles as well as the median. The whiskers show the minimum and maximum values inside 1.5∗IQR. Outliers are displayed as points outside of the whiskers.

Next, we focus on another important variable: the completion time. Throughout our experiment, the participants were allowed to submit their code as soon as they wanted. In Fig. 4, we display the distribution of completion times in all treatments. Without performance incentives, the participants spent on average 16 min and 22 s on the experiment. Implementing OSIT decreased the time to 12 min and 25 s (Wilcoxon-Mann-Whitney test, p-value = 0.170, padjusted = 0.262). In contrast, in the MPIT treatment, participants spent more time (20 min and 39 s, Wilcoxon-Mann-Whitney test, p-value = 0.131, padjusted = 0.262). We can further see in Fig. 4 that differently applied incentives (MPIT vs. OSIT) can lead to different levels of effort in terms of the time spent in the experiment (Wilcoxon-Mann-Whitney test, p-value = 0.005 padjusted = 0.015). In total, the differences in completion time are substantial between the treatments, even though they are not always statistically significant.

Figure 4 Distribution of the completion times.

The boxes show the 25% and 75% quantiles as well as the median. The whiskers show the minimum and maximum values inside 1.5∗IQR.

Using a post-experimental questionnaire, we further measured engagement, worry, and stress (cf. Fig. 5). In addition to the differences we can observe in these short scales, we also see that the self-reported engagement negatively correlates with completion times. This implies that participants who wanted to succeed in the task hurried. While the total sample sizes are again an issue, we observe some evidence that MPIT may have caused higher levels of engagement, distress, and worry, which is in line with the explanation through social pressure.

Figure 5 Self-reported values of engagement, distress, and worry.

The boxes show the 25% and 75% quantiles as well as the median. The whiskers show the minimum and maximum values inside 1.5∗IQR. Outliers are displayed as points outside of the whiskers.

Eye-tracking analysis

Approximately half of our participants in every treatment conducted the experiment with eye trackers. We can see no evidence that eye-tracking changed their performance (Wilcoxon-Mann-Whitney test, NPIT: p-value = 0.702 padjusted > 0.99), OSIT: p-value = 0.277, padjusted = 0.831, MPIT: p-value = 0.535, padjusted > 0.99). After evaluating the quality of the eye-tracking data, we had to exclude seven of 31 observations due to (1) low gaze detection (<70%) during the whole timespan or (2) high validation accuracy (>1.5 °) and high validation precision (>1 °) during the eye tracking calibration. This left us with 7/7/10 observations in NPIT/OSIT/MPIT, respectively. Still, the eye-tracking data provides us with valuable information on the participants’ behavior.

First, we split the lines with respect to their content into three blocks, that we define as Areas of Interest (AOI). We can see across all treatments that participants focused more on the second AOI, which includes the code of the sorting algorithm (cf. AOI 2 in Fig. 1). This section includes a nested for-loop and is, therefore, arguably the most complex section to analyze in our whole example. Second, we can observe a strong negative correlation between fixations (normalized to completion time) and F1-score. This indicates that participants who refocused on different gaze points more often had lower F1-scores, which may be interesting for further eye-tracking-based research in software engineering. The average fixation duration for participants in OSIT (300.32 ms) is lower compared to both NPIT (356.44 ms) and MPIT (334.58 ms), but is again not significant (OSIT-NPIT: p-value = 0.228, padjusted = 0.456, OSIT-MPIT: p-value = 0.406, padjusted = 0.812). This indicates that participants in OSIT spent less time focusing on one specific gaze point. Participants in OSIT also had the highest number of fixations normalized to completion time ( x¯NPIT=2.46, x¯OSIT=2.76, x¯MPIT=2.70), which could indicate that the time constraints led to more but shorter fixations.

Summary

In total, our results indicate that different financial incentives can alter participants’ behavior in software-engineering experiments, sometimes in less predictable ways. Surprisingly, the F1-score was the highest for NPIT. However, it remains arguable whether the F1-score is the best measure since we observe different relations between our incentive structures and different performance measures. We further recognize the completion time as a relevant measure, for which we could see that it can be predicted by the incentive structure and self-reported engagement. Simultaneously, the completion time seems to be a good predictor for the F1-score. We further stress that it would have been helpful to have a bigger sample size since our current sample size allows only very large effect sizes (Cohen’s d > 1.16) to become statistically significant.

Discussion

In this section, we discuss our key results in light of further literature in software engineering and experimental economics. First, we focus on the results from our survey. Second, we address our findings from the pre-registered results of our experiment. Finally, we discuss our exploratory results.

Software engineers like bonuses based on (Company) performance

Our survey results indicate that the most commonly applied payment scheme (i.e., fixed wages) does not have any performance-dependent components. However, several survey participants indicated that their employer applies bonuses dependent on company performance (i.e., team-dependent bonuses). Further, the results indicate that a substantial amount of software engineers would prefer performance-dependent incentives of different types. This finding is in line with what Beecham et al. (2008) report in their systematic literature review on the motivation in software engineering. Precisely, Beecham et al. (2008) indicate that increased pay and benefits that are linked to performance are among the factors that motivate software developers. Still, we cannot observe a clear picture from our results whether a specific component dominates all others. The MP component is a company bonus, a common element of total wages that is known to have positive effects on performance (Bloom & Reenen, 2011; Friebel et al., 2017; Garbers & Konradt, 2014; Guay, Kepler & Tsui, 2019). Similarly, by investigating successful IT organizations’ human resource practices, Agarwal & Ferratt (2002) found that providing bonuses as monetary rewards is among the practices employed to retain the best IT talent. As the number of participants in our survey was comparatively small, we cannot derive meaningful statistics from these numbers. Nonetheless, our results are a hint that software engineers wish for such elements to be implemented and that they are potentially sensitive to them.

Designing financial incentives is hard, but they have an impact on different variables

From our results, we can observe substantial differences in several important variables used in software-engineering experimentation, such as the time participants spend on a task or the number of bugs found/missed. These differences are meaningful in their impact on the interpretation of experimental results. Yet, since we preregistered the F1-score as our main dependent variable and obtained only a small sample size, the statistical analysis of treatment effects on the F1-score does not indicate significant results. We note that the treatment effect works in the other direction than we hypothesized (cf. “Laboratory Experiment”): Subjects without performance incentives (NPIT) had a higher F1-score than in MPIT or OSIT. Since this contrasts with the majority of economics literature, we now discuss possible explanations.

First, researchers have observed that financial incentives can have detrimental effects (Gneezy, Meier & Rey-Biel, 2011). Yet, this usually can only occur if the extrinsic motivational effect of the incentives is not strong enough to outweigh potential losses in intrinsic motivation. This is not a likely explanation for our experiment, in which the participants earned 23.83 € on average within a mean duration of 16.5 min. Such a payoff is substantially higher than the average hourly wage for student assistants at the university of 12 € per hour. Participants not being sensitive to such financial incentives would imply a very high a priori intrinsic motivation of the participants to conduct our experiment, which seems implausible.

Second, it is unclear whether the F1-score is the best metric for such analyses. Literature in economics usually does not make use of F1-scores. Instead, it focuses on the effect of incentives on context-specific criteria (e.g., number of hours worked, number of tasks solved, revenue, profit). However, research on the role of financial incentives on performance in software engineering is scarce. So, we applied a widely used, generic performance measure, the F1-score. Looking at other metrics that we captured, we do see some typical changes in performance despite our low numbers of observations. For example, it is in line with classical economics theory (Holmstrom & Milgrom, 1991) and empirical findings (Hong et al., 2018; Lazear, 2000) that in a multidimensional problem (e.g., quality and time) humans adjust towards the incentivized dimension. In this context, it means that when time is costly, people would optimize for it and speed up. This implies that the completion times in OSIT should be lower than in the other treatments, which is what we observed. Further, speeding up can easily lead to overlooking bugs (FN), which we also observed. These findings are also in line with the results of other software-engineering experiments conducted with students. Within their controlled experiment on requirements reviews and test-case development (Mäntylä et al., 2014) found that time pressure led to a decrease in the number of defects detected per time unit. In another experiment on manual testing, Mäntylä & Itkonen (2013) also observed a decreased number of defects detected per time unit due to time pressure. Our findings also align with developers’ behavior in real-life settings, in which short release cycles can lead to developers trading quality for completing tasks on time. For instance, an exploratory survey by Storey, Houck & Zimmermann (2022) at Microsoft revealed that developers are more likely to consider productivity in terms of the number of tasks completed in a given period and trade quality for quantity. Lastly, our eye-tracking data further supports that time pressure was perceived by the participants and changed their behavior. For instance, they had more fixations, but at shorter average fixation duration when facing time pressure.

Finally, note that, especially for OSIT, it is a very complicated process to induce value in line with real-world incentives (of open-source developers). Open-source developers can fall in a large variety of motivation schemes, including those being paid for their work independent of success and those working on the projects without any payment. In fact, the motivations of open-source developers are mostly intrinsic or internalized, such as reputation, learning, intellectual stimulation, altruism, kinship (e.g., desire to work in development teams), and belief that source code should be open (Gerosa et al., 2021; Bitzer, Schrettl & Schröder, 2007). The findings of a large-scale survey by Gerosa et al. (2021) point out that, in addition to all these intrinsic factors, career development is also relevant to many open-source software contributors as an extrinsic motivator. In our experiment, we aimed to rebuild the incentives for open-source developers who are not getting paid by companies and whose major incentive is to make things work (e.g., to help other people). The way we induced this incentive scheme via a payoff function (i.e., a large value for achieving the goal, a penalty for the time used) can cause some participants to not even try to find all bugs—since finding all bugs may be unrealistic and time-consuming (i.e., costly). Still, this very issue is similar to the real-life case of open source software development, where for a single individual, it may be too unrealistic to achieve the goal alone. This may imply that on the individual level, such incentives in fact induce a worse performance than a flat payment and the effectiveness of open source software engineering comes from a large number of contributors and not from the efficiency of the individual incentives. This would be a very interesting perspective for an experiment, yet would also require a much larger number of observations.

Eye trackers do not threaten the experimental design

Fourth, concerning eye-tracking, we measured that our participants spent most time on the nested for-loop of our code example. This is highly plausible, since cognitive complexity (Campbell, 2018) is relatively high in this part of our example. Importantly, with our setup, we did not measure any effects of having eye-trackers on participants’ measurable performance. This implies that eye-trackers pose no threats to the validity of an experiment. However, this result should be considered with caution, due to the low number of observations. Consequently, we strongly suggest to conduct future studies on this matter.

Threats to validity

In this section, we discuss possible threats to the validity of our study. Overall, our primary study design represented a typical controlled experiment in the lab, which improves the internal validity to increase the trust that any differences between the groups are due to the incentivization schemes we used. Still, the following threats to the internal and external validity remain.

Internal validity

Our study faces some potential threats concerning the choice of the code-review task, the incentives, and the dependent variable, which first impact internal validity, but can also expand to the external validity. First, our code-review task had to be designed in a way that is solvable for the participants of the experiment. Otherwise, we could not observe the additional effort induced by the incentives through any performance metric. We designed our task and thereby reduced this threat by conducting a pilot study with a different group of students. The results of that pilot indicated that our task can be solved by the students, but still required effort to solve (cf. “Laboratory Experiment”). The argument that the task was demanding but solvable is further supported by our actual experimental data, in which we can see that only two subjects were able to find all the bugs. This, however, was mostly due to bug number 2, which was the hardest to spot. The other bugs were easier to find, meaning that, for a substantial amount of participants, performance depended on effort.

Second, for incentives to work, they have to fulfill three criteria: monotonicity, salience, and dominance (Smith, 1976). Our experiment fulfills all these criteria as the incentives used (i.e., money) fulfill the criteria that participants prefer more of the incentive over less (monotonicity). The incentives were also salient, meaning that participants were informed how their decisions would influence their payoff. Moreover, the size of our payoffs is higher than the average hourly wage for student assistants, which we can take as a benchmark because it motivates typical students to work (dominance). So, we argue that we mitigated this threat to the internal validity as far as possible.

Lastly, the metric we chose to measure is another concern regarding internal validity. Specifically, it is unclear whether the F1-score is the best metric for such an experiment. In the data, we can observe that even in cases where the F1-score stays similar, other metrics (e.g., TPs or time spent on the task) can vary. However, a priori there was no indication against choosing the F1-score since it is quite an objective performance metric that weights between different types of true and false assessments. Consequently, future experiments with a different set of metrics can provide further insights into the impact of financial incentives. Still, our results provide valuable insights and already indicate how financial incentives can be used, also guiding the design of future experiments on the matter.

Looking at the average profits of the participants indicates another potential threat. Due to the different incentivization schemes, there are significant differences regarding the average payoffs between treatments (NPIT: 30.00 €, OSIT: 14.61 €, MPIT: 26.74 €, p < 0.0001). Yet, note that this is neither a threat to internal validity nor an explanation for performance differences. Specifically, it is not the average size of the realized payoff that is important for the incentivization, but the a priori saliently presented structure. For example, for OSIT, we observed the lowest average payoffs. However, this is the treatment with the highest possible payoff (up to 46.80 €, as compared to a maximum of 32.50 €/30.00 € for MPIT/NPIT). This in itself is another indicator that it is not solely about the size of the incentives, but also about their structure that matters to motivate participants.

External validity

Concerning external validity, the chosen task represents a typical exercise for practitioners. It is evident that a single code-review task cannot depict the whole variety of tasks in the real world, yet it represents a meaningful example. Another perspective is the choice of participants in our study. The participants in our experiment were mostly students. We are aware of ongoing debates on the comparability between student and professional participants (Höst, Regnell & Wohlin, 2000; Falessi et al., 2017). Therefore, the generalizability of our experiment towards practice may be more limited compared to conducting it with professional developers. Yet, such an alternative experiment would result in severely higher costs (due to paying practitioners instead of students).

Next, we focus on the external validity of the treatments we designed. The incentives in NPIT and MPIT are related to practice, since they have occurred prominently in our survey. In contrast, we designed the incentives for OSIT based on existing research and personal experiences with open-source development to depict one specific type of open-source project. Other researchers may have come up with different incentive schemes. However, for the chosen type of project, for which it matters to achieve a certain goal, the chosen incentives are realistic. Moreover, even if other payoff functions would have been more realistic or appropriate, this does not threaten the goal of our experiment to compare how different incentives impact participants’ performance. Our functions were different enough to achieve this goal, and we actually revealed performance differences.

A last threat to the external validity concerns the representativity of our survey. This survey was important to obtain information on possible incentive schemes in practice. To achieve the best results, it would have been best to conduct a large-scale, representative survey. In contrast, our survey is based on a convenience sample of mostly men, which may introduce biases (Zabel & Otto, 2021). Thus, the survey cannot provide generalizable results, including, but not limited to, the incentive schemes desired by women in software engineering (Otto et al., 2022). To increase the sample size, we interviewed eight practitioners from one company, which further limits the representativity and generalizability of the results. This, in turn, can imply a threat to the validity of the incentive schemes we designed. For instance, if the MP incentives from our survey are not the same as those of a more general sample of developers, the measured effects are less comparable to the real world. Yet, we mitigated this threat by checking for differences in responses from the three sub-samples, and we did not observe such differences. Also, again, our schemes were different enough to nonetheless reason on their impact on the performance of participants in software-engineering experiments.

Conclusion

In this article, we reported the results of a preregistered study (Krüger et al., 2022). We investigated in how far financial incentives impact the performance of (student) participants in software-engineering experiments. Doing so, we first surveyed the most commonly applied and preferred incentive schemes, and then implemented these in a laboratory experiment. Despite a low sample size, we observed strong effects of different incentives concerning variables like the time participants spent on their tasks or the number of correctly identified bugs. Yet, we did not observe significant differences concerning the F1-score as our primary metric. In addition, we used an eye-tracking analysis to investigate how the participants reviewed the code. Our findings indicate that participants correctly identified the most complex part of the code and spent the largest share of time on it. Further, our results indicate no performance differences between participants with or without eye-tracking, which supports the use of eye-tracking in future software-engineering studies. As the key message of our study, we found that software-engineering experiments are impacted by how participants are incentivized. How to design incentives to motivate the “ideal” behavior is a challenging task, though. Our contributions provide guidance in doing so, serving as exemplars and pointing out challenges researchers may face in this context.

Our results imply several opportunities for future work. First, different organizations may have different perspectives on the weight of different types of errors (software in healthcare vs. entertainment). This leads to the question of whether organizations in these domains apply different types of incentives. Second, there may be differences between the weights of errors between employers/managers and employees. For instance, do managers think that certain performance schemes induce more effort while the employees think otherwise? Research on this intersection of economics, psychology, and software engineering topics would highly benefit the understanding of the effects of incentives in software engineering.

Additional Information and Declarations

Competing Interests

The authors declare that they have no competing interests.

Author Contributions

Dmitri Bershadskyy conceived and designed the experiments, performed the experiments, analyzed the data, performed the computation work, prepared figures and/or tables, authored or reviewed drafts of the article, and approved the final draft.

Jacob Krüger conceived and designed the experiments, analyzed the data, performed the computation work, prepared figures and/or tables, authored or reviewed drafts of the article, and approved the final draft.

Gül Calıklı conceived and designed the experiments, analyzed the data, performed the computation work, prepared figures and/or tables, authored or reviewed drafts of the article, and approved the final draft.

Siegmar Otto conceived and designed the experiments, authored or reviewed drafts of the article, and approved the final draft.

Sarah Zabel conceived and designed the experiments, performed the experiments, analyzed the data, prepared figures and/or tables, authored or reviewed drafts of the article, and approved the final draft.

Jannik Greif performed the experiments, analyzed the data, performed the computation work, prepared figures and/or tables, authored or reviewed drafts of the article, and approved the final draft.

Robert Heyer conceived and designed the experiments, authored or reviewed drafts of the article, and approved the final draft.

Ethics

The following information was supplied relating to ethical approvals (i.e., approving body and any reference numbers):

The local Ethics Review Board of the Department for Mathematics and Computer Science at Eindhoven University of Technology, The Netherlands, approved the study on October 24, 2022 (ERB2022MCS21).

Data Availability

The following information was supplied regarding data availability:

All of our experimental objects and anonymized results are available on OSF: Zabel, Sarah, Dmitri Bershadskyy, and Siegmar Otto. 2024. “A Laboratory Experiment on Using Different Financial-Incentivisation Schemes in Software-Engineering Experimentation.” OSF. September 16. https://osf.io/mcxed/.

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
