# Peer review of "A laboratory experiment on using different financial-incentivization schemes in software-engineering experimentation"

_PeerJ Computer Science, doi:10.7717/peerj-cs.2650_

## Round 0.1 · accepted · Accept

One can clearly see that your paper has already gone through the preregistration phase with great reviews. Both reviewers and I agree that your paper fulfills all expected standards and is ready for publication.

There are a few suggestions for minor improvements in the text from the reviewers that you will find helpful for the final submission of the manuscript. Yet, there is nothing major that would warrant another official revision.

·

Basic reporting

no comment

Experimental design

no comment

Validity of the findings

no comment

Additional comments

I understand that the manuscript has already undergone a successful pre-registration phase and has received a public review from an independent expert who recommends the study for publication. Thus, I approach the manuscript with interest rather than a critical stance. After reading, I can affirm that a critical perspective was not necessary: the study's design and reporting are very good. The authors discuss aspects of their study with a level of scrutiny that even surpasses my own. For instance, they frequently address the small sample size, which indeed likely limited the study’s statistical power from the outset, preventing statistically significant effects from emerging. Nevertheless, the study still offers substantial insights, and considering the use of eye-tracking in the experiment, the sample size aligns reasonably well with the standards in software engineering research.

Several aspects stand out positively, such as the discussion section, where the authors contextualize their findings within existing literature. This section offers interested readers additional value. The publication of the data and the general trend toward pre-registrations are also commendable.

I have three minor suggestions for improvement or correction:

1. Supplemental Materials: In the supplemental file "03_data_survey.xlsx," I was curious about the meanings of the numbers in each cell. I assumed I would find an explanation in the README file, but I was unable to open it (I attempted with text and markdown editors, as well as directly on the online platform: https://osf.io/heabv?view_only=602088776ce5498597c473e74bbe0394).

2. Discussion of Validity Threats: When reading the study design, I already expected that the authors would address the potential validity threat of having mostly student participants in the discussion. I would, however, have appreciated a deeper exploration of which specific aspects of being a student (e.g., age, potentially correlated with life experience and attitudes towards incentive structures [citation]) the authors believe might influence the generalizability of the results, either according to their own opinion or existing literature. I lack the expertise to judge this case specifically, but the discussion currently seems to downplay a potentially impactful confounding variable. If the authors have more insights on this topic, it could enhance the discussion and provide more guidance for future studies.

3. Typos:
- Page 10: "... from software engineering Abid et al. (2019) to explore" should likely be "... from software engineering (Abid et al., 2019) to explore."
- Page 10: "Note that we could [not?] identify any impact wearing eye-trackers may have had on our participants during our analysis."

·

Basic reporting

no comment

Experimental design

the experimental design is well described and the authors faithfully admit various possible shortcomings to their sampling procedure or general threats to validity. As for table 1, I would recommend a forward reference to "the assumptions" mentioned in column "analysis plan" (later indicated to be for normality), and to explain why H2 is a hypothesis about inequality (whereas most statistical test assume equality of quantities and from that derive an "expected" distribution of the test statistics; starting from an inequality hypothesis is a bit unusual). Also, for perhaps better readability of the design, I would suggest letting the acronyms that appear as a footnote on page 4 come earlier in the regular text (on page 3 already so that one has them in mind already when seeing the table).

Validity of the findings

Given the relatively small sample size, the data did not lend itself to statistically significant claims, and the authors admit this. So, they let the data speak for itself, which I perceive as correct here.

Additional comments

Concerning the threats to validity (end of section 3.1), I think the authors admitted faithfully a potential sampling bias, but their argument could be strengthened perhaps from telling how many different countries “appear” in the sample (to underpin their line). Other than that, I did not understand the purpose of the study to find “the best financial motivator”, but rather to study the effect of “financial incentives to participate in an experiment”; in this respect, I think the actual form of the reward plays less of a role, since it is the “existence” of the reward that is to be tested. Thus, I think the threats to validity are actually less than the authors may think.

Along the same lines, however, I did not fully understand why the authors were asking people about their (subjectively) preferred incentive mechanisms (based on performance or non performance). The introduction and related work gave me the impression of the research gap to be how well financial incentives aid the conduction of experiments, but what I could tell (as a practitioner of software engineering) would perhaps only be the incentive to “perform” in my job about software engineering (also if I were an open source software developer). So, the experimental design seems perfectly well aligned, but appears to rather address different aspects or dimensions of the incentive mechanisms (comparing different versions thereof). To avoid a false impression during the introductory parts already, I would suggest adding a line explaining that the paper does not only study whether or not a financial incentive would be effective, but rather being more specific about what incentive (from a list of candidates) would be the most useful. If this were stated as the goal, the design is sure without question.

External reviews were received for this submission. These reviews were used by the Editor when they made their decision, and can be downloaded below.